# Multimodal Fusion for Trust Assessment in Lower-Limb Rehabilitation: Measurement Through EEG and Questionnaires Integrated by Fuzzy Logic

**DOI:** 10.3390/s25216611

**Published:** 2025-10-27

**Authors:** Kangjie Zheng, Fred Han, Cenwei Li

**Affiliations:** School of Automation and Intelligent Manufacturing, Southern University of Science and Technology, Shenzhen 518055, China; 12331366@mail.sustech.edu.cn (K.Z.); 12433034@mail.sustech.edu.cn (C.L.)

**Keywords:** trust assessment, lower-limb rehabilitation, EEG, questionnaires, fuzzy logic

## Abstract

This study aimed to evaluate the effectiveness of a multimodal trust assessment approach that integrated electroencephalography (EEG) and self-report questionnaires compared with unimodal methods within the context of lower-limb rehabilitation training. Twenty-one mobility-impaired participants performed tasks using handrails, walkers, and stairs. Synchronized EEG, questionnaire, and behavioral data were collected. EEG trust scores were derived from the alpha-beta power ratio, while subjective trust was assessed via questionnaire. An adaptive neuro-fuzzy inference system was used to fuse these into a composite score. Analyses included variance, correlation, and classification consistency against behavioral ground. Results showed that EEG-based scores had higher dynamic sensitivity (Spearman’s ρ = 0.55) but greater dispersion (Kruskal–Wallis H-test: *p* = 0.001). Questionnaires were more stable but less temporally precise (ρ = 0.40). The fused method achieved stronger behavioral correlation (ρ = 0.59) and higher classification consistency (κ = 0.69). Cases with discordant unimodal results revealed complementary strengths: EEG captured real-time neural states despite motion artifacts, while questionnaires offered contextual insight prone to bias. Multimodal fusion through fuzzy logic mitigated the limitations of isolated assessment methods. These preliminary findings support integrated measures for adaptive rehabilitation monitoring, though further research with a larger cohort is needed due to the small sample size.

## 1. Introduction

Trust is defined as the willingness to accept vulnerability based on confident expectations of another’s behavior [1]. Within the context of human–robot interaction (HRI) in rehabilitation, this translates to a patient’s confidence in the capability and reliability of a specific assistive device to safely support them during therapeutic activities [2]. Patients must trust both the healthcare providers and rehabilitation devices—such as walkers and canes—to actively engage in therapeutic activities. Insufficient trust in assistive devices can increase gait variability and prolong recovery [3], while excessive trust may cause over-reliance and injury risks [4]. Accurate trust assessment is therefore essential for tailoring rehabilitation interventions to patients’ subjective experiences and physical engagement. However, current assessment tools fail to capture the dual interpersonal and human–machine dimensions of trust in lower-limb rehabilitation, underscoring the need for specialized, context-sensitive evaluation methods.

Trust assessment plays a crucial role in lower-limb rehabilitation by enabling optimized therapeutic engagement and outcomes. Effective evaluation necessitates the integration of both subjective experiences and physiological signals to dynamically capture trust fluctuations. As trust represents not merely a cognitive state but a complex construct governed by behavioral and biological factors through intricate neurophysiological interactions [5,6]. Real-time monitoring allows clinicians to detect immediate confidence shifts and adapt interventions accordingly [7], while trust level classification (e.g., low, moderate, high) facilitates personalized treatment planning [8]. This integrated approach provides comprehensive insights into trust dynamics that are essential for successful rehabilitation.

Subjective assessment tools, particularly self-report questionnaires, offer valuable insights into users’ subjective states and have been widely used to evaluate interpersonal and human–computer trust. Established instruments such as the Rotter Interpersonal Trust Scale and customized scales capture rich contextual and experiential data [9]. However, these tools are inherently limited to post hoc evaluations and are susceptible to biases such as social desirability [10]. For instance, patients may report inflated trust ratings despite demonstrating behavioral hesitancy, creating misalignment between self-reported and observable trust indicators [11,12].

Objective physiological measures address these limitations through the continuous monitoring of nervous system activity [5]. EEG, in particular, offers the potential for the real-time, continuous measurement of neural correlates associated with trust during rehabilitation (e.g., P300) and frequency-domain features (e.g., frontal alpha asymmetry and theta/beta ratio). The alpha-to-beta power ratio (α/β ratio) is capable given its specificity in representing cognitive load, emotional calmness, and physical relaxation [13]. Studies have demonstrated that heightened alpha power is often associated with relaxation, reduced anxiety, and focused attention, while elevated beta power reflects active engagement, vigilance, and cognitive effort in trust-related interactive tasks [5,14]. In motor rehabilitation, where patients simultaneously process physical movements and trust-related decisions, the α/β ratio offers interpretative advantages due to its sensitivity to the global activation state of the central nervous system [7].

However, despite its high temporal resolution and sensitivity to dynamic changes, EEG data can be influenced by non-cognitive factors, which may interfere with trust measurements [15]. Muscle artifacts, for example, may generate signals that mimic neural oscillations, potentially resulting in an overestimation of trust-related neural activity. Heightened alertness or anxiety (indicated by increased beta wave activity) can be misinterpreted as low trust, even in cases where patients hold a positive subjective attitude toward the intervention [16]. Therefore, an accurate trust assessment through EEG requires careful consideration of the patient’s actual experiences and potential sources of artifact.

Behavioral metrics offer a complementary approach, providing objective benchmarks for validating trust assessments. Observable behaviors, including adherence rates, decision-making latency, device dependency, and intervention frequency, are less susceptible to reporting biases [17]. Researchers have effectively utilized behavioral metrics as the standard for evaluating physiological and subjective trust indicators [18], recognizing that behaviors during rehabilitation ultimately reflect trust levels.

To overcome the limitations of unimodal approaches, researchers have explored multimodal data fusion techniques. Combining peripheral physiological signals, such as electrodermal activity (EDA) and electrocardiogram (ECG), has been shown to enhance the accuracy and robustness of trust assessments [18,19]. Moreover, fuzzy logic methods offer unique advantages in trust assessment due to their ability to handle uncertainty, manage nonlinear relationships between variables, and integrate heterogeneous data types [7]. Furthermore, the use of minimally invasive physiological parameters (e.g., electrocardiogram, respiration) within data-driven fuzzy logic algorithms has shown promise in estimating dynamic psychophysiological states such as attention, fatigue, and stress [20]. Therefore, fuzzy logic can accommodate the inherent ambiguity and subjectivity in trust assessments, making it ideal for integrating physiological and subjective data.

However, there is a significant gap in the literature regarding the integration of central nervous system signals (e.g., EEG) with subjective measures (e.g., questionnaires) in a complementary manner, especially within the context of lower-limb rehabilitation [21]. Due to the complex interplay of cognitive trust and physical engagement in this setting, such integrated methods have great potential.

Therefore, this study proposes a multimodal approach to assess situation-specific patient trust in lower-limb rehabilitation devices by integrating EEG-derived physiological signals with questionnaire-based reports. This study systematically compared this integrated method against unimodal approaches (EEG or questionnaires alone) to develop a comprehensive trust assessment system through synergistic signal integration. It is hypothesized that a fused trust score combining EEG and questionnaire data will demonstrate stronger correlations with behavioral outcomes and greater classification consistency than individual measures. An experimental protocol incorporating typical rehabilitation scenarios (e.g., assisted walking, stair climbing) was designed to collect synchronized multimodal data. A fuzzy logic system was developed to integrate neural and subjective trust indicators, with statistical analyses conducted to validate the fused measure against behavioral ground truth.

## 2. Materials and Methods

### 2.1. Experimental Setup

The rehabilitation training scenario was designed to simulate authentic challenges commonly encountered in lower-limb rehabilitation while also systematically eliciting varying degrees of trust in the rehabilitation process, specifically trust in the rehabilitation devices being used. The experimental protocol incorporated several commonly used assistive devices: a single-point cane, a wide-based walker, a climbing stair, and a handrail-equipped path (Figure 1).

To experimentally manipulate trust levels during the rehabilitation training, three distinct scenarios were implemented, each designed with carefully calibrated levels of difficulty (see Figure 2):Single-point cane and wide-based walker: Participants navigated a path with obstacles of varying sizes placed along both sides. To increase the perceived challenge and assess trust under controlled conditions, paper and glass cups were placed on top of obstacles, requiring careful negotiation to avoid contact.Climbing stair: Training intensity was gradually increased by progressively shortening the allowed time to complete the task as participants gained more confidence—this enabled observation of their reliance on the equipment under increased physical demands.Handrail-equipped path: Participants navigated a handrail-equipped path with low-height obstacles placed along its length. This setup required precise stepping and balance control, challenging their trust in their own abilities and the support provided by the handrails.

### 2.2. Participants

Twenty-one participants (Mean Age = 73, SD = 6.3) aged 60 to 85 years with lower-limb mobility impairment were recruited from two nursing homes in Zhongshan and Pingshan cities in China. Inclusion criteria included ability to walk without a wheelchair; walking with the help of canes; and able to complete stair climbing. Exclusion criteria included ability to walk without any assistance; requiring a caregiver to supervise while walking with assistive devices; and unable to communicate with researchers or caregivers. Prior to participation, lower limb mobility was assessed by a 10 m walk test to ensure that the participants could safely complete all experimental activities. All participants provided written informed consent after a thorough explanation of the study’s purpose, procedures, and potential risks. Ethical approval for this study was obtained from the Ethics Committee of Southern University of Science and Technology (project identification code: 20250070).

### 2.3. Data Acquisition

#### 2.3.1. Questionnaire

Trust assessment was conducted using a questionnaire adapted from established trust scales [22]. The questionnaire consisted of 12 items (Table 1) and assessed the participants’ trust in the assistive device and rehabilitation process. Responses were recorded using a ten-point Likert scale (1 = Strongly Disagree, 10 = Strongly Agree). To minimize bias, researchers who administered the questionnaire were blinded to the participants’ EEG performance. Each questionnaire item was explicitly linked to specific training events and behavioral milestones (e.g., crossing an obstacle, ascending stairs, navigating around an obstacle) to ensure temporally precise alignment between the subjective trust reports and physiological measurements across the different training phases.

#### 2.3.2. EEG

A wireless, wearable EEG sensor (RunE W2, EEG-Pro) mounted on a headband was used to record brain activity during the rehabilitation training. The EEG device records EEG signals at a sampling frequency of 500 Hz. Electrode-skin contact impedance was maintained below 10 kΩ throughout the experiment to ensure signal quality. The sensor was positioned on the participants’ foreheads to minimize interference with their movements. The alpha/beta power ratio was extracted from the EEG data, which is a validated neural correlate of trust [14]. To control for individual baseline differences, a 10 s resting-state recording was obtained before each session [23]. Continuous EEG data collection enabled the standardization of task-related alpha/beta ratios using z-score transformation relative to this resting baseline [24], quantifying trust-related neural dynamics relative to individual reference states.

#### 2.3.3. Behavior

Rehabilitation behaviors were recorded using a video camera (1080p, 30 fps) to capture full-body movements and interactions within structured training scenarios. This behavioral data served as a baseline for comparison and validation with the EEG and questionnaire measures. Based on the trust measurement framework proposed by Campagna et al. (2025), four key behavioral measures were employed: adherence rate, decision latency, device dependency, and intervention frequency [17,25]. The operational definitions of each measure are shown in Table 2.

To convert observed behaviors into quantifiable trust or performance scores, a structured scoring process was implemented:Define behavior: Clearly define the behavior (e.g., intervention frequency) based on the indicators in Table 2;Set the observation context: Standardize the observation context around a specific task (e.g., walking up and down five stairs);Record behavior frequency: Record the frequency of the behavior (e.g., adjusting the grip on the handle) within a fixed task unit or time interval;Evaluate performance: Convert the frequency counts and overall performance into a scale score using a predefined scoring scale (e.g., 0–10);Calculate and interpret the score: The final score is interpreted as reflecting the user’s level of trust or adaptability (e.g., fewer interventions and successful task completion indicate higher trust, while frequent corrections or task failures indicate lower trust).

### 2.4. Data Processing and Analysis

#### 2.4.1. Unimodal Data Processing

Questionnaire data processing:

The internal consistency of the questionnaire was assessed using Cronbach’s alpha (α), calculated based on the inter-item correlations between all questions in the trust-related scale. The calculated α value was 0.82, indicating good reliability and exceeding the traditional threshold of 0.70 widely accepted in the field [26].

To compare the questionnaire with EEG and behavioral dynamics, the questionnaire responses were temporally aligned with specific behavioral events during the rehabilitation training. This synchronization method, based on timestamps, adheres to established practices for aligning multimodal data in behavioral research [27]. First, the participants were asked to recall specific training programs and share their experiences. Second, the responses were aligned with the training programs during questionnaire analysis. Finally, the questionnaire responses were annotated with specific video clips using ELAN 6.9 video analysis software. This process ensured accurate temporal correspondence between each subjective answer and the observed behavioral event.

The aligned data were then used to generate continuous time–trust curves for each participant, reflecting changes in trust over time. These trust curves served as the basis for subsequent quantitative and qualitative analyses.

EEG data processing:

The EEG data processing pipeline was implemented in Python 3.12 and consisted of the following steps:

Data import and artifact removal: Raw EEG data were imported and preprocessed to enhance signal quality. A zero-phase bandpass filter (1–45 Hz) was first applied to eliminate slow drifts and high-frequency noise. Subsequently, a hybrid artifact removal strategy combining wavelet transformation (Daubechies 4 wavelet, 5 decomposition levels with soft thresholding) and median filtering (kernel size = 5) was implemented to effectively suppress physiological and motion artifacts [28,29]. The efficacy of this processing pipeline is demonstrated in Figure 3, which displays the synchronized EEG traces from Participant 1 across four rehabilitation scenarios (resting, handrail walking, stair climbing, walker walking). Successful synchronization of EEG with specific rehabilitation tasks was achieved through timestamp alignment with video recordings, ensuring precise temporal correspondence between the neural signals and behavioral contexts.

Frequency band decomposition: Five classic frequency bands were extracted using finite impulse response (FIR) filters designed with the firwin function in MNE-Python: delta (1–4 Hz), theta (4–8 Hz), alpha (8–13 Hz), beta (13–30 Hz), and gamma (30–45 Hz). Although the study focused on the alpha and beta bands for trust score estimation, data from all five classical frequency bands were extracted. This comprehensive approach aligns with standard EEG processing pipelines to ensure methodological completeness. Furthermore, monitoring activity in all bands’ aids in data quality control, for instance, by identifying artifacts prevalent in specific bands like delta.Resting-state baseline establishment: A continuous 10 s segment from the beginning of the recording was used as the resting-state baseline. The alpha–beta power ratio was calculated from this segment using Welch’s periodogram method and used as a reference for subsequent normalization of task-related trust scores.Task period analysis: The task period was segmented into 3 s epochs (from −1 to +2 s around decision points), with non-overlapping windows. Band power within each epoch was computed using Welch’s method with a 2 s window length (nperseg = 1000 samples at 500 Hz sampling rate) and default 50% overlap.Trust scores were derived through Z-score normalization followed by hyperbolic tangent mapping. Specifically, the alpha–beta power ratios across all task epochs were first standardized into Z-scores

Z = (X − μ)/σ,(1)
where μ and σ represent the mean and standard deviation of all ratios. These Z-scores were then transformed using the equation:Trust Score = 5 + 2.5 × tanh(Z),(2)
mapping them onto a physiologically interpretable 1–10 scale [23]. The parameter selection for this transformation adhered to general principles of psychophysical scale design [30]. The offset value of 5 ensured that the mean of the data (Z = 0) corresponded to the midpoint of the scale, thereby providing a symmetric scoring range. The scaling factor of 2.5 was chosen to optimize the dynamic output range for the majority of the expected data (Z ≈ ±2) to approximately 3–8. This design effectively mitigates extreme floor and ceiling effects [31], ensuring discriminative validity across the scale and facilitating subsequent analysis.

Behavior data processing:

Behavioral data were coded and scored using ELAN video analysis software. To minimize scoring bias, two independent researchers, blinded to the participants’ EEG data and questionnaire responses, conducted parallel behavioral assessments. Inter-rater reliability was rigorously quantified using Cohen’s kappa (κ). A Cohen’s kappa value of 0.85 indicated strong agreement between the researchers. Discrepancies were addressed through collaborative discussion and consensus, referencing the operational definitions of the behavioral indicators.

#### 2.4.2. Multimodal Data Fusion by Fuzzy Logic Method

To address the arbitrariness of manual rule design, an adaptive neuro-fuzzy inference system (ANFIS) was employed to integrate EEG-based and questionnaire-based trust scores, using behavioral trust scores as the ground truth for supervision [32]. Given the limited sample size (N = 21), the model was initialized parsimoniously [33] with two inputs—normalized EEG and questionnaire scores—each assigned two Gaussian fuzzy sets (‘Low’ and ‘High’). This formed a minimal rule base covering all input combinations:Rule 1: If (EEG score is low) and (questionnaire score is low) then (output = p_1_)Rule 2: If (EEG score is low) and (questionnaire score is high) then (output = p_2_)Rule 3: If (EEG score is high) and (questionnaire score is low) then (output = p_3_)Rule 4: If (EEG score is high) and (questionnaire score is high) then (output = p_4_)
where p_1_ to p_4_ are linear functions of the inputs. This grid-based rule initialization provided a comprehensive and unbiased starting point for data-driven optimization [34].

The hybrid learning algorithm subsequently adjusted both the membership functions and the consequent parameters, evolving the initial rules into a refined model. To evaluate performance and prevent overfitting, leave-one-out cross-validation (LOOCV) was applied [35], yielding a root mean square error (RMSE) of 1.17 and a determination coefficient (R^2^) of 0.78, indicating good performance.

Finally, a model trained on the full dataset produced the input–output surface in Figure 4, visually representing the data-derived nonlinear fusion of neurophysiological and subjective trust measures.

#### 2.4.3. Method for Trust Dynamics Assessment

To evaluate variability and interrelationships among the trust measurements, a multifaceted statistical approach was adopted. Prompted by the need to account for the specific characteristics of the dataset including questionnaire, EEG, and behavior, the Shapiro–Wilk test was used to assess the data distribution of each modality. The results showed that the EEG data of multiple samples deviated from normality at *p* < 0.05. This may be associated with artifacts, nonlinear characteristics, and complexity inherent in electroencephalographic signals. Therefore, this finding guided the selection of non-parametric methods for subsequent analyses.

For the variance analysis, the non-parametric Kruskal–Wallis H test was designed as the primary method for comparing the dispersion of trust scores, given its robustness to non-normal data [36]. In parallel, a traditional ANOVA was designated as a complementary analysis. The congruence between the results of both tests would serve to strengthen the validity of our findings. Significant results were followed by Dunn’s post hoc tests with Bonferroni correction to identify differing modality pairs. For correlations, a triangulation approach using Pearson’s r, Spearman’s ρ, and Kendall’s τ was employed to holistically assess both linear and monotonic associations, ensuring conclusions on validity were not assumption-dependent. This strategy ensures that conclusions regarding convergent validity and measurement robustness are not dependent on a single set of statistical assumptions.

#### 2.4.4. Method for Trust Level Classification

To evaluate the agreement in trust level classification across modalities, Cohen’s kappa coefficient was used to compare EEG-based, questionnaire-based, and fused trust classifications against behavioral classifications as the reference standard. This analysis assessed the consistency and accuracy of each modality in categorizing trust levels.

Additionally, a confusion matrix was constructed to visualize classification discrepancies between the EEG and questionnaire outcomes. Representative cases with divergent classifications were further analyzed by comparing EEG traces, questionnaire responses, and behavioral performance to identify potential sources of inconsistency such as physiological artifacts, response biases, or task engagement effects.

## 3. Results

### 3.1. Analysis of Trust Dynamics Assessment

#### 3.1.1. Variance Analysis of Trust Assessment

To compare the variability of trust level measurements across different assessment modalities, the Kruskal–Wallis H test was employed as the primary statistical method due to the non-normal distribution of our data. This analysis revealed a statistically significant difference in the dispersion of scores across the four modalities, H (3) = 16.8, *p* = 0.001. The effect size was large (ε^2^ = 0.19), indicating that the modality type accounted for a substantial proportion of the variability in trust scores.

Post hoc Dunn’s tests with Bonferroni correction were conducted to pinpoint the specific differences. The pairwise comparisons confirmed that the dispersion of EEG-derived trust scores was significantly greater than that of both the behavioral scores (*p* = 0.012) and the questionnaire scores (*p* = 0.001). No other pairwise comparisons reached statistical significance.

This primary non-parametric finding is further supported by the descriptive variances presented in Table 3. The pattern of variances—with EEG scores showing the highest variance (σ^2^ = 5.31), followed by behavioral (σ^2^ = 3.66), fused (σ^2^ = 3.88), and questionnaire scores showing the lowest (σ^2^ = 1.93)—aligned perfectly with the results of the Kruskal–Wallis H test and the subsequent post hoc analysis. This consistent pattern across both non-parametric hypothesis testing and descriptive statistics strengthens the validity of the observed differences in score dispersion among the assessment modalities.

#### 3.1.2. Correlation Analysis of Trust Assessment

To examine the relationships between trust levels measured through different modalities, we performed a comprehensive correlation analysis using a triad of coefficients: Spearman’s ρ (to capture monotonic relationships), Kendall’s τ (as a robust non-parametric measure), and Pearson’s r (to capture linear relationships). This multi-method approach was adopted to ensure the robustness of our findings, irrespective of the specific distributional characteristics of the data. The results are summarized in Table 4.

Consistent with the variability analysis, the correlation patterns revealed clear hierarchical associations. Questionnaire scores demonstrated weak-to-moderate correlations with the behavioral scores (Spearman’s ρ = 0.40, Kendall’s τ = 0.31, Pearson’s r = 0.40,). In contrast, the EEG-derived trust scores showed consistently stronger associations with behavioral outcomes across all correlation coefficients (Spearman’s ρ = 0.55, Kendall’s τ = 0.43, Pearson’s r = 0.58). Notably, the fused scores, which integrated EEG and questionnaire data, achieved the highest correlation magnitudes with the behavioral benchmark (Spearman’s ρ = 0.59, Kendall’s τ = 0.44, Pearson’s r = 0.64).

### 3.2. Analysis of Trust Level Classification

To assess the consistency of trust level classification across different measurement modalities, Cohen’s kappa coefficient was calculated to compare the agreement between EEG-based, questionnaire-based, and fused trust assessments with behavioral classification results (Table 5). The kappa values for the questionnaire (κ = 0.51) and EEG (κ = 0.49) were comparable, indicating moderate agreement with the behavioral trust classification [37]. In contrast, the fused measurement of EEG and questionnaire achieved a higher kappa value (κ = 0.69).

To statistically determine whether the agreement strength differed between modalities, pairwise comparisons of the kappa coefficients were performed. After applying a Bonferroni correction for three comparisons, the fused method demonstrated significantly superior agreement with behavioral measures compared with both the questionnaire method (*p* = 0.02) and the EEG method (*p* = 0.03). No significant difference was found between the questionnaire and EEG methods (*p*= 0.72).

Furthermore, a confusion matrix was constructed to compare the classification discrepancies between the EEG and questionnaire results (Figure 5). The normalized confusion matrix revealed distinct patterns. When the EEG classification was ‘High’ (12 cases), it most frequently coincided with a ‘High’ questionnaire score (66.7%, 8 cases), while discrepancies with ‘Middle’ and ‘Low’ questionnaire scores were less common (16.7% each, 2 cases). For cases with a ‘Middle’ EEG classification (4 cases), the outcomes were split evenly between the ‘High’ and ‘Middle’ questionnaire scores (50.0% each, 2 cases), with no instances of a ‘Low’ questionnaire score. Conversely, when the EEG classification was ‘Low’ (5 cases), it showed a diverse association with questionnaire results: the most common correspondence was with the ‘Middle’ and ‘Low’ questionnaire scores (40.0% each, 2 cases), followed by a ‘High’ questionnaire score (20.0%, 1 case).

One case from each of these scenarios was randomly selected, and the EEG, questionnaire, and behavioral trust scores were plotted to analyze their inconsistencies, as shown in Figure 6.

High EEG/low questionnaire: Participant P14 had a high EEG confidence score (8/10), but a low questionnaire score (4/10) and moderate behavioral performance (5/10). The questionnaire was more consistent with the observed behavior, with the participant demonstrating cautious movements (e.g., total time to complete the movement 385 s) and self-reported activity limitations (e.g., “I moved slowly due to poor lower limb control”). However, the discrepancy in the high EEG score may be due to the following. (1) Physiological inhibition: reduced global neural activity due to physical weakness may mask anxiety-related signals. In contrast, motor dysfunction may reduce sensorimotor beta wave activity. (2) EMG contamination: Artifacts from frontalis muscle activity may inflate EEG-derived confidence scores by mimicking high-frequency, low-amplitude neural oscillations.High EEG score/medium questionnaire score: For example, Participant 1’s EEG score (9/10) was closer to his behavioral response (7.5/10) than to his neutral questionnaire response (5/10). This participant’s questionnaire responses frequently included neutral and ambiguous expressions such as “acceptable” and “indifferent”. This suggests that an ambiguous interpretation of the questions may be the reason why questionnaire scores do not reflect the actual behavior.Middle EEG /low questionnaire: Participant 15’s EEG (6/10) and behavioral scores (6.5/10) converged. However, on the questionnaire (4/10), the participant reported dissatisfaction with the assistive device (e.g., “the cane felt uncomfortable”). Notably, this subjective discomfort did not affect the actual task performance or EEG patterns, highlighting the disconnect between fleeting usability complaints and sustained trust.Middle EEG/high questionnaire: Participant 19 completed the training program quickly (174 s) and reported high confidence (8/10), but had a moderate EEG score (5/10). Observation of the participant’s behavior revealed prolonged gaze fixation on their feet during the training session, indicating a high level of focus on the task. This may explain the suppressed alpha power and elevated beta power, indicating a high level of focus on the training task, but not reflecting the participant’s overall trust level.Low EEG/high questionnaire: Participant 17′s EEG (4/10) and behavior (4.5/10) suggested low trust, conflicting with overly positive self-reports (“very easy”, “I am healthy”). This may reflect social desirability bias, wherein the participants overstated competence to conform to perceived expectations.

## 4. Discussion

This study examined the performance of questionnaire-based, EEG-based, and fused multimodal approaches in assessing both dynamic trust levels and trust classification outcomes within a lower-limb rehabilitation training context. By comparing each method’s results against ground-truth behavioral measures through variance analysis, correlation assessment, and consistency evaluation, their respective capabilities in trust measurement were systematically evaluated.

Furthermore, through in-depth analysis of discordant cases between the questionnaire and EEG assessments, the strengths and limitations of each modality were elucidated, and the complementary potential of multimodal integration was explored. These findings provide valuable methodological insights and practical guidance for trust assessment in lower-limb rehabilitation training.

### 4.1. EEG Demonstrates High Sensitivity to Dynamic Trust but Remains Vulnerable to Physiological Confounds

The results demonstrated that EEG exhibits high sensitivity in capturing immediate fluctuations in trust, as evidenced by its stronger correlation with behavioral trust indicators (Spearman’s ρ = 0.55) compared with questionnaires (ρ = 0.40). This indicates that EEG can detect rapid changes in trust that static assessment methods may overlook. However, the significant dispersion in EEG data (H-test: *p* = 0.001), along with misclassification cases (such as inflated P14 scores caused by α/β shifts from muscle artifacts), revealed the susceptibility of EEG to physiological confounds. These artifacts as well as the potential for cognitive states such as hypervigilance (observed at P19) to be mistaken for genuine trust reflect the challenges encountered in the practical measurement of trust using EEG, as discussed by Campagna et al. (2023) [16].

Although EEG offers distinct advantages in real-time trust assessment for rehabilitation training, its practical application requires the implementation of robust artifact rejection techniques and the careful consideration of potential cognitive confounds. Multidimensional information calibration, incorporating detailed behavioral observations and contextual information, is essential to prevent bias in phased rehabilitation training assessments. This approach allows for disambiguating neural signatures specifically related to trust from confounding factors, providing more reliable information to support rehabilitation training decisions.

### 4.2. Questionnaires Provide Contextual Stability but Lack Temporal Resolution

In contrast to the dynamic sensitivity of EEG, questionnaires offer a rich subjective experience that is crucial for contextualized trust assessment. Their unique strength lies in their ability to capture nuanced personal perspectives that neural measures alone cannot fully capture. For instance, Participant P15 reported discomfort with assistive devices, a factor that, while not affecting immediate task performance, could potentially impact long-term trust formation and rehabilitation adherence. These findings align with research suggesting that self-reported trust levels can fluctuate based on the performance of external entities (such as assistive technology), thereby influencing rehabilitation outcomes.

However, questionnaires are susceptible to inherent biases that can compromise the accuracy of trust assessments. For example, Participant P17′s self-reports, influenced by social expectations (“very simple”, “I am healthy”), were inconsistent with their actual rehabilitation behaviors. Social expectations and perceived norms can significantly affect how participants perceive and describe their relationships with healthcare providers and assistive technologies, influencing the accuracy of self-reported trust levels. This finding is consistent with research from Leichtmann and Nitsch (2021) and Wang (2022), highlighting the significance of considering interpersonal dynamics and social desirability in trust assessment within healthcare contexts [10,12].

Furthermore, questionnaires exhibit limited temporal sensitivity in capturing dynamic trust fluctuations, as evidenced by the weak behavioral correlation (Spearman’s ρ = 0.40). This indicates that questionnaires are more suitable for assessing stable trust traits and general attitudes rather than capturing transient, real-time changes in trust. Therefore, when employing questionnaires for trust assessment, it is crucial to focus on the trust context, social relationships, and individual experiences and assess and calibrate for potential biases systematically. Strategies such as employing validated social desirability scales (e.g., the Marlowe–Crowne Social Desirability Scale) [38] and carefully framing questions to minimize bias (e.g., using neutral wording, avoiding leading language, and embedding lie scales within instruments) are essential [39].

### 4.3. Multimodal Fusion Optimizes the Sensitivity-Specificity Trade-Off

The fusion of EEG and questionnaire data achieved superior performance compared with unimodal approaches, demonstrating the benefits of integrating subjective and physiological measures. The fused method produced the highest behavioral correlation (Spearman’s ρ = 0.59) and classification consistency (κ = 0.69), indicating a synergistic effect that mitigated the limitations inherent in the individual modalities. The temporal sensitivity of EEG compensated for the limitations in capturing the temporal dynamics of trust through questionnaire measurements, as evidenced by the increase in Spearman correlation between questionnaire trust dynamics and behavior from 0.40 to 0.59 after fusion with EEG data.

Conversely, subjective data obtained from the questionnaire, including trust propensity, personal experience, and social factors, contributed to reducing the risk of EEG being misled by physiological signals. For instance, the inclusion of self-reported action constraints corrected the EEG overestimation observed in Participant P14.

However, the findings also highlight the need for the cross-validation of self-reported information with objective physiological data to ensure its validity. As demonstrated in the case of P17, social expectations can mask an individual’s genuine trust experience in self-reports, necessitating the integration of objective EEG information to determine their trust status accurately. Therefore, effective fusion strategies may need to be tailored to the different stages of recovery and the specific type of self-reported information being considered.

### 4.4. Implications for Trust Assessment in Lower-Limb Rehabilitation

Implement a dynamic calibration mechanism: A dynamic calibration mechanism is crucial for accurate and reliable trust assessment. Leveraging the advantages of multimodal assessment fusion allows for integrating the real-time assessment capabilities of EEG with the rich contextual information gathered from questionnaires, enabling cross-validation and continuous calibration of different modalities. This involves the cross-validation and calibration of different modalities throughout the rehabilitation process, requiring the collection of detailed information on the trust context including physical conditions, experience with assistive devices, social support, and the participant–therapist relationship. A one-time calibration may not be sufficient to capture the evolving nature of trust; therefore, multiple trust assessment calibration iterations should be performed throughout training to provide the most accurate information for rehabilitation training decisions.Construct a context-aware multimodal weighting framework: A context-aware multimodal weighting framework is essential for optimizing the integration of different assessment elements. This involves identifying specific trust contexts and assigning appropriate weights to various data streams. For example, when participants are in good physical condition and feel empowered, they may be more likely to express their genuine wishes and concerns [40]. In such cases, the weight of the questionnaire should be appropriately adjusted to minimize the influence of social expectations.Linking trust to adaptive robotic control and rehabilitation outcomes: The trust assessment framework developed in this study can be directly translated into an input for adaptive control systems in robotic rehabilitation and assistance devices. By establishing a real-time, closed-loop system, the fused trust score (derived from EEG and subjective reports) can inform the adjustment of key robotic control parameters. For instance, when a lower-limb exoskeleton robot assists a patient in completing sit-to-stand rehabilitation training, as the patient’s trust increases, the system can gradually reduce the level of assistance provided by the device, thereby encouraging active participation and promoting neuroplasticity. Conversely, detecting a decrease in trust may trigger an increase in guiding force or a decrease in movement speed to enhance the patient’s sense of security and stability. This approach aligns with recent research analyzing how patient participation correlates with variations in impedance control parameters [41].Furthermore, beyond robotic control, trust levels should be linked to broader rehabilitation outcomes (e.g., adherence, functional gains, and satisfaction). This allows clinicians to identify critical trust thresholds and implement proactive measures. For example, early warning systems for declining trust can trigger personalized interventions such as motivational interviewing or adjustments to the training protocol. These advances contribute to the development of precision rehabilitation frameworks that dynamically adapt both the robotic assistance and the therapeutic strategy according to the individual’s evolving psychophysiological state.

By integrating the insights gained from this study, clinicians and researchers can develop more effective trust assessment and intervention strategies to improve outcomes for individuals undergoing lower-limb rehabilitation.

## 5. Conclusions

This study provides compelling evidence that a multimodal trust assessment approach, integrating EEG and self-report questionnaires via fuzzy logic, significantly outperforms unimodal approaches in capturing participant trust dynamics during lower-limb rehabilitation training. The enhanced performance of the fused model, as evidenced by its higher correlation with behavioral outcomes and improved classification consistency, underscores its practical reliability and precision in predicting real-world rehabilitation behaviors. Although EEG signals are susceptible to motion artifacts, they offer a unique advantage in capturing dynamic, real-time sensitivity to trust fluctuations. Conversely, questionnaires provide consistent and contextually rich subjective insights; however, the possibility of bias influences these advantages. The integration of these modalities mitigates individual limitations, forming a more comprehensive and robust representation of participant trust. These findings support the combined use of psychophysiological and subjective indicators within rehabilitation settings. This integrated approach can enable more adaptive monitoring strategies, personalized intervention protocols, and ultimately, improved participant outcomes.

Notwithstanding these promising results, several limitations of this study should be acknowledged. The most significant limitation is the relatively small sample size (N = 21). While sufficient for an initial proof-of-concept and model development, this limitation introduces several potential biases and constraints on the generalizability of our findings. The small sample size increases the risk of overfitting and may limit the detection of smaller but potentially meaningful effects. Furthermore, it constrains the demographic and clinical diversity of the cohort, which could affect the external validity of the results.

Future studies should prioritize expanding the sample size to enhance the robustness and external validity of these results. Specifically, employing advanced validation techniques such as cross-validation in larger, more diverse cohorts is recommended to obtain a more reliable estimate of model performance. Furthermore, multi-center, longitudinal studies that recruit larger and more diverse rehabilitation populations (e.g., patients with stroke, spinal cord injury, or osteoarthritis) should be initiated to validate the generalizability of the trust assessment framework and investigate how trust evolves over extended rehabilitation periods.

## Figures and Tables

**Figure 1 sensors-25-06611-f001:**
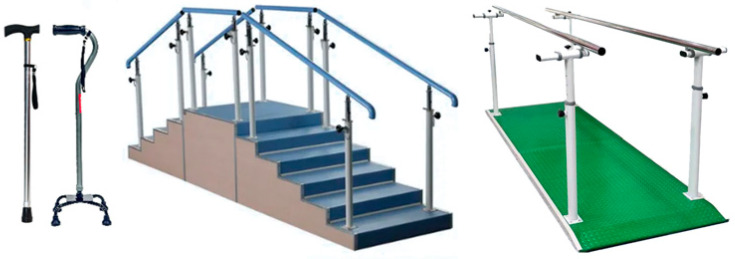
Lower-limb rehabilitation assistive devices.

**Figure 2 sensors-25-06611-f002:**
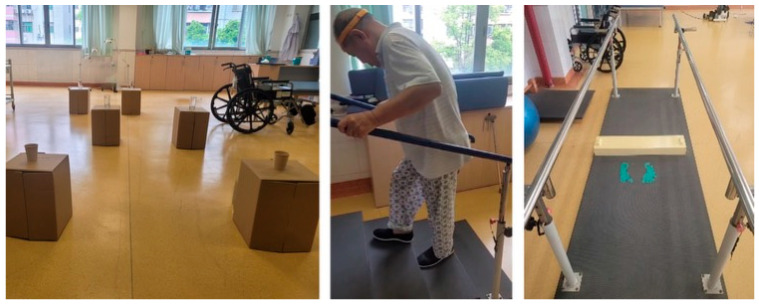
Lower-limb rehabilitation training scenarios.

**Figure 3 sensors-25-06611-f003:**
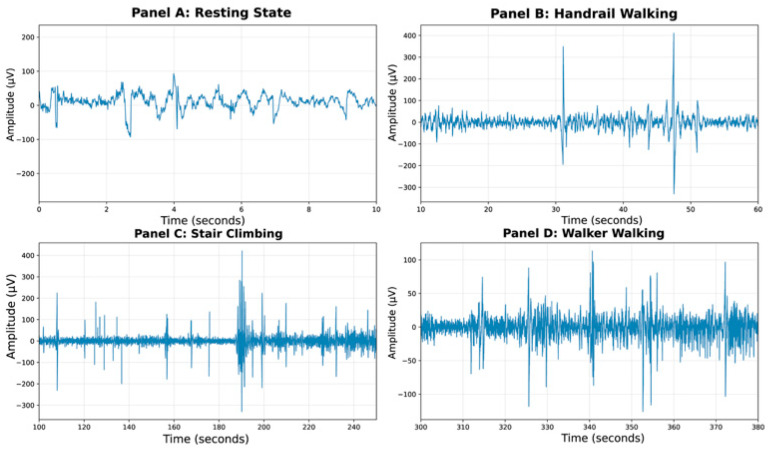
EEG signals across different rehabilitation situations.

**Figure 4 sensors-25-06611-f004:**
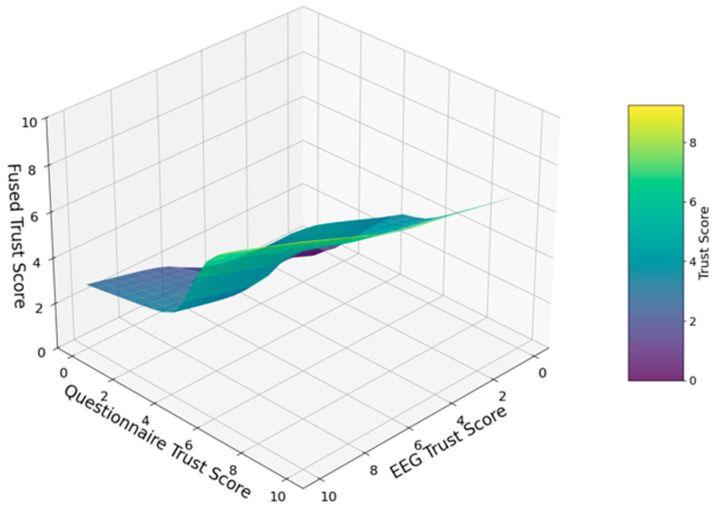
ANFIS-based trust score fusion surface.

**Figure 5 sensors-25-06611-f005:**
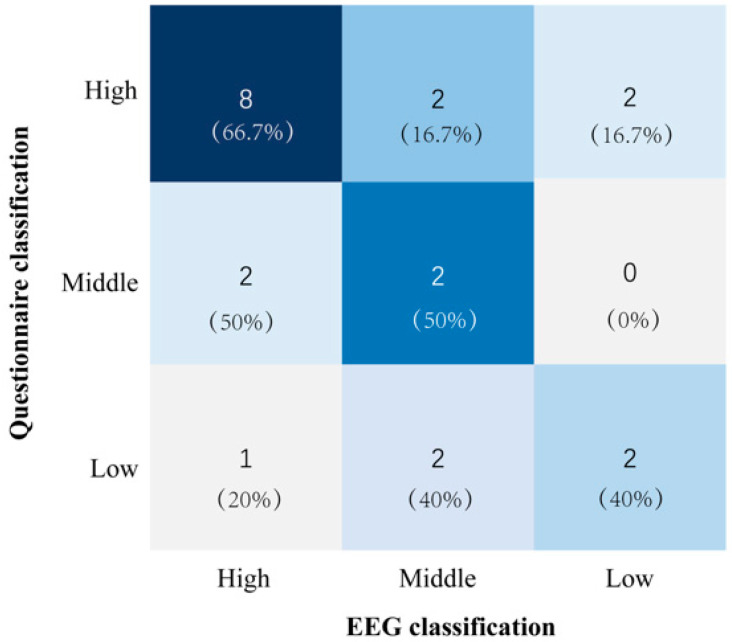
Confusion matrix of the trust classification results of the questionnaire and EEG.

**Figure 6 sensors-25-06611-f006:**
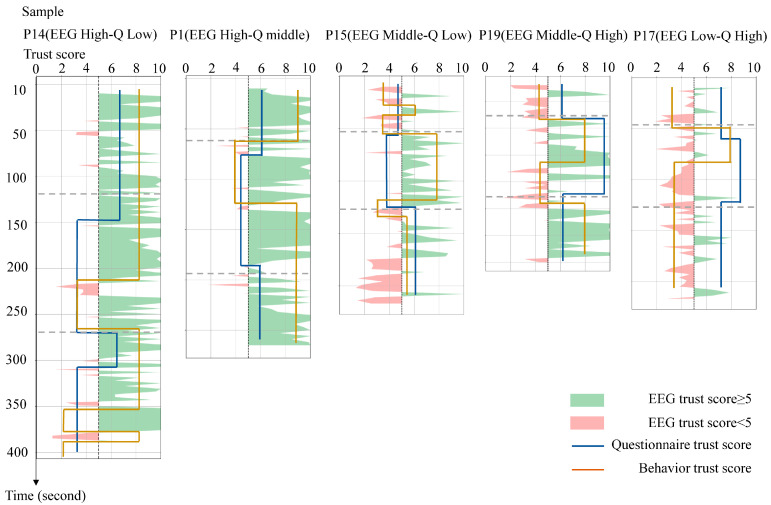
Case of inconsistency between the questionnaire and EEG trust assessment results.

**Table 1 sensors-25-06611-t001:** Trust questionnaire.

Item Number	Questions
1	I believe that there could be negative consequences when using the rehabilitation device.
2	I feel I must be cautious when using the rehabilitation device.
3	It is risky to interact with the rehabilitation device.
4	I believe that the rehabilitation device will act in my best interest.
5	I believe that the rehabilitation device will do its best to help me if I need help.
6	I believe that the rehabilitation device is interested in understanding my needs and preferences.
7	I think that the rehabilitation device is competent and effective in its role.
8	I think that the rehabilitation device performs its role as a rehabilitation assistant very well.
9	I believe that the rehabilitation device has all the functionalities I would expect from it.
10	If I use the rehabilitation device, I think I would be able to depend on it completely.
11	I can always rely on the rehabilitation device for my training.
12	I can trust the information presented to me by the rehabilitation device.

**Table 2 sensors-25-06611-t002:** Behavioral observation indicators.

Indicator	Sub-Indicator	Description
Compliance	Instruction adherence	Percentage of rehabilitation commands correctly executed (e.g., stepping up a step, going around an obstacle).
Decision time	Decision making	Time taken to make the decision to initiate an action (e.g., time from obstacle/stair recognition to action decision-making).
Reliance	Device reliance level	Frequency and extent of using physical help or guidance from the device (e.g., cane, handrails).
Intervention	Intervention frequency	Frequency of attempts to modify, correct, or pause device operation (e.g., adjust the height of the crutches, change the way gripping the handles, and adjust body balance).
Verification	Active verification	Frequency of additional visual or physical verification behaviors (e.g., looking at device components, touching handrails, or scanning obstacles).

**Table 3 sensors-25-06611-t003:** Variance of single-modality and multi-modality trust measurement data.

EEG	Questionnaire	Fused Method	Behavior
5.31	1.93	3.38	3.66

**Table 4 sensors-25-06611-t004:** Correlation coefficients (Pearson’s r, Spearman’s ρ, Kendall’s τ) between the trust assessment modalities (questionnaire, EEG, fused) and the behavioral trust scores (N = 21).

Method	Spearman	Kendall	Pearson
Questionnaire	0.40	0.31	0.40
EEG	0.55	0.43	0.58
Fused	0.59	0.44	0.64

**Table 5 sensors-25-06611-t005:** Agreement (Cohen’s kappa) of trust level classification between the questionnaire, EEG, and fused modalities against the behavioral benchmark (N = 21).

Method	Kappa	*p* Value
Questionnaire	0.51	0.002
EEG	0.49	0.015
Fused	0.69	0.010

## Data Availability

The data presented in this study are available upon request from the corresponding author. The data are not publicly available due to privacy reasons.

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
