# Peer review of "Multimodal Fusion for Trust Assessment in Lower-Limb Rehabilitation: Measurement Through EEG and Questionnaires Integrated by Fuzzy Logic"

_sensors, 2025, doi:10.3390/s25216611_

Round 1

Reviewer 1 Report

Comments and Suggestions for Authors

The main weaknesses of the manuscript is that the manuscript highly relies on statistical results in a small sample set, and both parametric and non-parametric tests have been used without justification. This should be clarified for the reader to trust the observations and conclusions.

I suggest to shorten the introduction.

Section 2.3.2. Please add the technical specifications of the recording device to allow the reader to properly assess the underlying quality of the data. E.g. which sampling rate was used ?

Section 2.4.1. Why were information in all five frequency bands extracted, if only the alpha and beta activity was used for estimating a trust score ?

Please add traces of EEG data from different situations to either the methods or results section to allow the reader to judge the signal quality and variability.

Section 2.4.3. ANOVA and Person’s r are parametric tests and therefore assume that the data are following a normal distribution.  Spearman’s and Kendall’s are non-parametric tests. The authors need to justify their choice of methodology is appropriate to use in relation to the author’s data set.

Section 3.1.1.

Fisher’s F-test also assume data follow a normal distribution – as above, the authors need to justify their choice of methodology is appropriate to use.

Table 4. The table caption should inform the reader which groups of parameters are tested, i.e. what is the questionnaire, eeg and fused tested against ?

Table 5. the table caption should inform the reader what the three parameters (questionnaire, eeg and fused) are tested against ?

Author Response

Dear reviewers,

We are very grateful for reviewing the paper so carefully. Your review comments are very helpful to our research. We have tried our best to improve and made some changes in the manuscript. Modifications are marked in blue color in the manuscript.

Comment 1:” The main weaknesses of the manuscript is that the manuscript highly relies on statistical results in a small sample set, and both parametric and non-parametric tests have been used without justification. This should be clarified for the reader to trust the observations and conclusions.”

Response: We appreciate your suggestions, and we have implemented them.

  1. Regarding the small sample size:

The reviewer rightly identifies sample size as an important consideration. We have explicitly acknowledged this limitation in the Results section, noting that while sufficient for initial proof-of-concept, the sample size (N=21) may affect generalizability and increase overfitting risk. We have further elaborated on future research directions, including the need for larger, multi-center longitudinal studies with more diverse patient populations to validate and extend our findings. These comprehensive revisions significantly strengthen the methodological rigor of our work and provide readers with clear justification for our analytical choices, enhancing confidence in the reported findings. (See Page 16 lines 579-593)

  1. Regarding the use of parametric and non-parametric tests:

We appreciate the reviewer's emphasis on methodological justification. To this end, we have significantly revised our statistical methods and reporting:

  • Added explicit justification: We now explicitly state in the Methods section (2.4.3) that "given the nonnormal distribution of the data, nonparametric analysis was primarily employed," using the Shapiro-Wilk test. This provides a clear rationale for our methodological choices. (See Page 9 lines 300-308)
  • Consistent nonparametric framework: We have systematically replaced parametric tests with nonparametric ones.
  • The Kruskal-Wallis H test was used as the primary method for comparing the dispersion of scores across modalities, while a traditional ANOVA was designated as a complementary analysis. (See Page9 309-319; Page 10 lines 333-350)
  • Spearman's ρ was used as the primary correlation measure and is reported consistently throughout the manuscript. The congruence between the results of both tests would serve to strengthen the validity of our findings. (See Page9 309-319; Page 10 lines 353-366)
  • Clarified multi-method : In 2.4.3, a triangulation approach using Pearson’s r, Spearman’s ρ, and Kendall’s τ was employed to holistically assess both linear and monotonic associations, ensuring conclusions regarding convergent validity and measurement robustness are not dependent on a single set of statistical assumptions. (See Page 9 lines 314-319)

Comment 2:” I suggest to shorten the introduction.”

Response: We thank the reviewer for this valuable suggestion. Accordingly, we have thoroughly revised the introduction by combining redundant sections and shortening the text to enhance readability. (See Page 2-3 line 44-116)

Comment 3:” Section 2.3.2. Please add the technical specifications of the recording device to allow the reader to properly assess the underlying quality of the data. E.g. which sampling rate was used?”

Response: We thank the reviewer for this suggestion. The requested technical specifications, specifically the sampling rate of 500 Hz, have now been added to Section 2.3.2 to allow for a proper assessment of data quality. (See page 5, lines 168-172)

Comment 4:” Section 2.4.1. Why were information in all five frequency bands extracted, if only the alpha and beta activity was used for estimating a trust score ?

Please add traces of EEG data from different situations to either the methods or results section to allow the reader to judge the signal quality and variability.”

Response: We are grateful to the reviewer for these valuable suggestions.

  1. Regarding the extraction of all five frequency bands. We have added a justification in Section 2.4.1. As revised, this comprehensive approach ensures methodological completeness and aids in data quality control, for instance, by helping to identify artifacts that may be prevalent in non-target bands like Delta. (See page 7, lines 240-245)
  2. Addition of EEG traces. Following the reviewer's suggestion, we have added a new Figure 3 to the Methods section (2.4.1). This figure presents synchronized EEG traces from different rehabilitation situations, enabling the reader to directly assess the signal quality and variability. (See page 7, lines 224-236; Figure 3)

Comment 5:” Section 2.4.3. ANOVA and Person’s r are parametric tests and therefore assume that the data are following a normal distribution. Spearman’s and Kendall’s are non-parametric tests. The authors need to justify their choice of methodology is appropriate to use in relation to the author’s data set.”

Response: We thank the reviewer for raising this critical methodological point regarding the assumptions of parametric tests. To rigorously address this, we have substantially revised Section 2.4.3.

Specifically, we first conducted a Shapiro-Wilk test to formally assess the normality of our multi-modal dataset. The results confirmed that key variables, particularly the EEG data, significantly deviated from a normal distribution (p < 0.05). This finding justified the primary use of non-parametric tests. (See page 9, lines 301–308)

Consequently, our revised analysis plan designates the Kruskal-Wallis H test as the primary method for variance analysis due to its robustness. Furthermore, we employed a triangulation approach for correlation analysis, using Spearman’s ρ and Kendall’s τ alongside Pearson’s r to ensure conclusions regarding convergent validity and measurement robustness are not dependent on a single set of statistical assumptions. (See page 9, lines 301–319; page 10, lines 333–366).

Comment 6:” Section 3.1.1.

Fisher’s F-test also assume data follow a normal distribution – as above, the authors need to justify their choice of methodology is appropriate to use.”

Response: We thank the reviewer for raising the critical point regarding the assumptions of parametric tests. In response to the comment that "Fisher’s F-test also assume data follow a normal distribution," we have removed the reference to the F-test from Section 3.1.1 to eliminate any methodological inconsistency.

The revised text now solely relies on and reinforces the primary non-parametric findings. We emphasize that the results of the Kruskal-Wallis H test and the subsequent post hoc Dunn's tests with Bonferroni correction are fully and consistently supported by the pattern of descriptive variances presented in Table 3. This adjustment ensures the statistical narrative is both robust and methodologically sound, based entirely on analyses appropriate for our data distribution. (see page 10 line 333-350)

Comment 7:” Table 4. The table caption should inform the reader which groups of parameters are tested, i.e. what is the questionnaire, eeg and fused tested against ?”

Response: We are grateful to the reviewer for pointing out the lack of clarity in the table caption. We have revised the caption of Table 4 to explicitly state the compared groups, as follows: "Correlations between trust scores derived from the questionnaire, EEG, and fused modalities, and the behavioral trust scores (N=21)."(See Page 10, lines 367-368).

Comment 8:” Table 5. the table caption should inform the reader what the three parameters (questionnaire, eeg and fused) are tested against ?”

Response: We are grateful to the reviewer for this comment. We have clarified the caption of Table 5 as suggested. It now reads: "Agreement (Cohen's Kappa) of trust level classification between the questionnaire, EEG, and fused modalities against the behavioral benchmark (N=21)." (See Page 11, lines 383-384).

Thank you for your valuable comments on our research. We hope that these revisions are sufficient to make our manuscript suitable for publication and look forward to hearing from you at your earliest convenience.

Reviewer 2 Report

Comments and Suggestions for Authors

This work presents the development and validation of a model for estimating patient trust in assistive devices using a population of 21 subjects. The approach relies on fuzzy logic to combine objective measures collected from EEG systems and questionnaires to capture subjective perception.

The work is very interesting, and this reviewer was impressed.

I would like to share a series of comments that could further improve the robustness of the presentation and the reproducibility of the results:
- The authors mention that the use of physiological parameters is an excellent indicator of central nervous system activity. The authors should extend their literature review to include works in which minimally invasive physiological parameters (collected with wearable sensors such as ECG, respiration, and EBG) have been used within data-driven algorithms based on fuzzy logic to estimate psychophysiological parameters such as attention, fatigue, stress, and energy expenditure. The process these authors are interested in—trust—is closely linked to the highly variable psychophysiological parameters during a rehabilitation session with exoskeleton robots.
- This reviewer does not believe that calling the questionnaires "psychological measures" is very accurate. The authors should check the literature to determine whether it is correct to call the subjective assessment of the user's perception a "psychological measure."
- The first lines of the Materials and Methods section (lines 133-138) are redundant with respect to what is stated at the end of the introduction. We suggest integrating all the concepts into the definition of the paper's objective at the end of the introduction.
- The scenarios are not presented in a consistent order between text and figures. The authors should make the discussion consistent.
- Did the authors obtain approval from an ethics committee to conduct the experiments? If so, they should include the approval code. If not necessary, the authors should clearly specify why this study is exempt.
- In general, this reviewer believes that the authors should specify throughout that the confidence they are trying to estimate is relative to the specific device or the situation. This is because if the authors are trying to estimate a continuous parameter that can dynamically change during an experience with the device, then this reviewer believes fuzzy logic is a very appropriate choice.
- The authors should clarify that the acquisition frequency is sampling the EEG signal.
- The authors should report the rules implemented in the fuzzy logic model.
- Increase the readability of Figure 3. The labels are too small.
- The confusion matrix can also be reported by normalizing by row and reporting percentages.
- The concept of modifying training intensity based on patient confidence is interesting. Recent studies have also analyzed how active participation may be related to variations in impedance control parameters during robotic rehabilitation sessions. The authors can provide insight into how the confidence estimation presented in this work can also provide input to the control parameters of robotic devices for rehabilitation and assistance.

Author Response

Dear reviewers,

We are very grateful for reviewing the paper so carefully. Your review comments are very helpful to our research. We have tried our best to improve and made some changes in the manuscript. Modifications are marked in blue color in the manuscript.

Comment 1:” The authors mention that the use of physiological parameters is an excellent indicator of central nervous system activity. The authors should extend their literature review to include works in which minimally invasive physiological parameters (collected with wearable sensors such as ECG, respiration, and EBG) have been used within data-driven algorithms based on fuzzy logic to estimate psychophysiological parameters such as attention, fatigue, stress, and energy expenditure. The process these authors are interested in—trust—is closely linked to the highly variable psychophysiological parameters during a rehabilitation session with exoskeleton robots.”

Response: We are grateful to the reviewer for this valuable comment. We agree that situating 'trust' within the framework of other dynamic psychophysiological parameters is crucial. Accordingly, we have expanded our literature review in the introduction to incorporate discussions on the use of wearable physiological sensors and fuzzy logic for estimating states like attention and fatigue, as referenced in the new citation [20]. This addition strengthens the foundation for our approach to trust assessment. (See Page 3 lines 94-99)

Comment 2:” This reviewer does not believe that calling the questionnaires "psychological measures" is very accurate. The authors should check the literature to determine whether it is correct to call the subjective assessment of the user's perception a "psychological measure.”

Response: We are grateful to the reviewer for pointing out the terminology inaccuracy. Upon checking the literature, we agree that "subjective assessment" more precisely describes the nature of the questionnaire-derived data, as it directly reflects the user's reported perception. We have therefore replaced "psychological measures" with "subjective assessment" across the manuscript to ensure terminological correctness. (See Page 2, line 54; page3, line101; page14, line 499)

Comment 3:” - The first lines of the Materials and Methods section (lines 133-138) are redundant with respect to what is stated at the end of the introduction. We suggest integrating all the concepts into the definition of the paper's objective at the end of the introduction.”

Response: We are grateful to the reviewer for pointing out the redundancy. We have followed the suggestion by integrating the relevant concepts into the introduction and streamlining the start of the Materials and Methods section. (See page 3, lines 105-116)

Comment 4:” The scenarios are not presented in a consistent order between text and figures. The authors should make the discussion consistent.”

Response: We appreciate the reviewer's careful reading. We apologize for the inconsistency and have now revised the manuscript to ensure the scenario order is consistent across all text and figures. (See page 3, lines 123-124; page 4, lines 130-141)

Comment 5:” - Did the authors obtain approval from an ethics committee to conduct the experiments? If so, they should include the approval code. If not necessary, the authors should clearly specify why this study is exempt.”

Response: We thank the reviewer for the comment. The ethics approval code (20250070) has been included in the Participants section (Section 2.2). (See page 4, lines 154-155)

Comment 6:” In general, this reviewer believes that the authors should specify throughout that the confidence they are trying to estimate is relative to the specific device or the situation. This is because if the authors are trying to estimate a continuous parameter that can dynamically change during an experience with the device, then this reviewer believes fuzzy logic is a very appropriate choice”

Response: We sincerely thank the reviewer for this critical insight. We fully agree that explicitly defining the scope of the assessed trust is essential for the clarity and precision of our work. In direct response to this comment, we have thoroughly revised the manuscript to specify that the trust being estimated is the patient's trust in the specific rehabilitation device within the specific training situation.

The key modifications are as follows:

1.In the introduction, immediately following the general definition of trust, we have added a sentence to anchor the concept within the context of human-device interaction in rehabilitation (Page 1, Lines 33-35).

2.The research objective at the end of the introduction has been rephrased to clearly state our aim of assessing "situation-specific patient trust in lower-limb rehabilitation devices" (Page 3, Lines 105-107).

3.In the Methods section (2.1 Experimental Setup), we have clarified that the experimental scenarios were designed to elicit varying trust "in the rehabilitation devices being used" (Page 3, Lines 121-122).

Comment 7:” - The authors should clarify that the acquisition frequency is sampling the EEG signal.”

Response: We thank the reviewer for this suggestion. The requested technical specifications, specifically the sampling rate of 500 Hz, have now been added to Section 2.3.2 to allow for a proper assessment of data quality. (See page 5, lines 168-172)

Comment 8:” The authors should report the rules implemented in the fuzzy logic model.”

Response: We thank the reviewer for prompting us to clarify the fuzzy logic rules. We have updated the manuscript as follows in Section 2.4.2:

1 Rule Specification: We now explicitly list the four initial fuzzy rules that were used to initialize the model.

2 Model Evolution: We clarify that these initial rules were not manually crafted but were used as a starting point. The ANFIS architecture then automatically learned and refined the model parameters (including the consequent linear functions p₁ to p₄ and the input membership functions) based on the behavioral trust scores. (See Page 8, lines 278-289).

Comment 9:” Increase the readability of Figure 3. The labels are too small.”

Response: We are grateful to the reviewer for this comment. We have Increased the readability of the Figure. (See Page 9, Figure 4). (Figure 4 is the original Figure 3, as a new picture was added.)

Comment 10:” The confusion matrix can also be reported by normalizing by row and reporting percentages.”

Response: We thank the reviewer for this suggestion. We have now included the normalized confusion matrix (as percentages) in Figure 5 and added a detailed description of the classification patterns in the manuscript, as shown in the revised text. (See Page 11, lines 385-395; Figure 5).

Comment 11:” The concept of modifying training intensity based on patient confidence is interesting. Recent studies have also analyzed how active participation may be related to variations in impedance control parameters during robotic rehabilitation sessions. The authors can provide insight into how the confidence estimation presented in this work can also provide input to the control parameters of robotic devices for rehabilitation and assistance.”

Response: We are grateful to the reviewer for this valuable suggestion and for highlighting the interesting connection to robotic control. As suggested, we have now provided a detailed discussion on how the trust estimation from our framework can be utilized to inform and adapt the control parameters of rehabilitation devices. This discussion, which includes specific examples like adjusting assistance levels in an exoskeleton, has been integrated into the manuscript and substantially enhances the application perspective of our study. (See Page 11, lines539-559).

Thank you for your valuable comments on our research. We hope that these revisions are sufficient to make our manuscript suitable for publication and look forward to hearing from you at your earliest convenience.

Reviewer 3 Report

Comments and Suggestions for Authors

The study shows multimodal EEG-questionnaire fusion via fuzzy logic improves trust assessment in lower-limb Rehabilitation. The Manuscript meets academic standards, but further revisions are needed.

1. The authors should provide more detail about the experimental setup, Like parameters for EEG data acquisition and processing.

2. The authors should consider performing additional statistical tests, such as pairwise comparisons with adjusted p-values for multiple testing. 

3. The discussion of limitations should be expanded to explicitly address potential biases introduced by the small sample size. And authors can add some suggestion about specific strategies for future research.

Author Response

Dear reviewers,

We are very grateful for reviewing the paper so carefully. Your review comments are very helpful to our research. We have tried our best to improve and made some changes in the manuscript. Modifications are marked in blue color in the manuscript.

Comment 1:” The authors should provide more detail about the experimental setup, Like parameters for EEG data acquisition and processing.”

Response: We thank the reviewer for suggesting that we provide more details about the experimental setup and EEG processing. In response, we have expanded Section 2.3.2 and the data processing pipeline to include specific parameters. The revisions now detail the EEG device model, sampling frequency, impedance control, as well as the specific bandpass filter range (1-45 Hz) and the complete parameters of our hybrid artifact removal strategy (utilizing a Daubechies 4 wavelet with 5 decomposition levels and a median filter with a kernel size of 5). We have also added Figure 3 to demonstrate the efficacy of this processing. (See Page5 lines 169-172; Page7 lines224-234; Figure3)

Comment 2:” The authors should consider performing additional statistical tests, such as pairwise comparisons with adjusted p-values for multiple testing.”

Response: We thank the reviewer for the important suggestion to perform pairwise comparisons with adjusted p-values for multiple testing.

We have thoroughly implemented this recommendation throughout our manuscript. Specifically:

In Section 3.1.1, the post hoc Dunn's tests following the significant Kruskal-Wallis test were conducted with Bonferroni correction for the pairwise comparisons between the four assessment modalities. (See Page 10, lines 339-343)

In Section 3.2, the pairwise comparisons of the Cohen's Kappa coefficients were also performed with a Bonferroni correction for the three pre-planned comparisons. (See Page 11, lines377-382)

Comment 3:” - The discussion of limitations should be expanded to explicitly address potential biases introduced by the small sample size. And authors can add some suggestion about specific strategies for future research.”

Response: We are grateful to the reviewer’s valuable suggestion to strengthen the discussion of our study's limitations and future directions. In direct response, we have substantially expanded the limitations to explicitly address the potential biases introduced by the small sample size (N=21). We now discuss its implications for overfitting, statistical power, and the generalizability of our findings due to constrained demographic and clinical diversity. (See page 16, lines 579-585)

Furthermore, we have added specific, actionable strategies for future research. These include employing advanced validation techniques like cross-validation in larger cohorts, and initiating multi-center, longitudinal studies to validate the framework's generalizability across diverse patient populations and to track the evolution of trust over time. We believe these revisions have significantly enhanced the depth and forward-looking perspective of our discussion. (See page 16, lines 586-593)

Thank you for your valuable comments on our research. We hope that these revisions are sufficient to make our manuscript suitable for publication and look forward to hearing from you at your earliest convenience.

Round 2

Reviewer 1 Report

Comments and Suggestions for Authors

Thanks for addressing the raised issues. No further comments. 

Reviewer 2 Report

Comments and Suggestions for Authors

The authors addressed all the comments raised in the previous round of review.